# DATA AUGMENTATION GENERATIVE ADVERSARIAL NETWORKS

## ABSTRACT

Effective training of neural networks requires much data. In the low-data regime, parameters are underdetermined, and learnt networks generalise poorly. Data Augmentation (Krizhevsky et al., 2012) alleviates this by using existing data more effectively. However standard data augmentation produces only limited plausible alternative data. Given there is potential to generate a much broader set of augmentations, we design and train a generative model to do data augmentation. The model, based on image conditional Generative Adversarial Networks, takes data from a source domain and learns to take any data item and generalise it to generate other within-class data items. As this generative process does not depend on the classes themselves, it can be applied to novel unseen classes of data. We show that a Data Augmentation Generative Adversarial Network (DAGAN) augments standard vanilla classifiers well. We also show a DAGAN can enhance few-shot learning systems such as Matching Networks. We demonstrate these approaches on Omniglot, on EMNIST having learnt the DAGAN on Omniglot, and VGG-Face data. In our experiments we can see over 13% increase in accuracy in the low-data regime experiments in Omniglot (from 69% to 82%), EMNIST (73.9% to 76%) and VGG-Face (4.5% to 12%); in Matching Networks for Omniglot we observe an increase of 0.5% (from 96.9% to 97.4%) and an increase of 1.8% in EMNIST (from 59.5% to 61.3%).

## 1 INTRODUCTION

Over the last decade Deep Neural Networks have produced unprecedented performance on a number of tasks, given sufficient data. They have been demonstrated in variety of domains (Gu et al., 2015) spanning from image classification (Krizhevsky et al., 2012; He et al., 2015a;b; 2016; Huang et al., 2016), machine translation (Wu et al., 2016), natural language processing (Gu et al., 2015), speech recognition (Hinton et al., 2012a), and synthesis (Wang et al., 2017), learning from human play (Clark & Storkey, 2015) and reinforcement learning (Mnih et al., 2015; Silver et al., 2016; Foerster et al., 2016; Van Hasselt et al., 2016; Gu et al., 2016) among others. In all cases, very large datasets have been utilized, or in the case of reinforcement learning, extensive play. In many realistic settings we need to achieve goals with limited datasets; in those case deep neural networks seem to fall short, overfitting on the training set and producing poor generalisation on the test set.

Techniques have been developed over the years to help combat overfitting such as dropout (Hinton et al., 2012b), batch normalization (Ioffe & Szegedy, 2015), batch renormalisation (Ioffe, 2017) or layer normalization (Ba et al., 2016). However in low data regimes, even these techniques fall short, since the the flexibility of the network is so high. These methods are not able to capitalise on known input invariances that might form good prior knowledge for informing the parameter learning.

It is also possible to generate more data from existing data by applying various transformations (Krizhevsky et al., 2012) to the original dataset. These transformations include random translations, rotations and flips as well as addition of Gaussian noise. Such methods capitalize on transformations that we know should not affect the class. This technique seems to be vital, not only for the low-data cases but for any size of dataset, in fact even models trained on some of the largest datasets such as Imagenet (Deng et al., 2009) can benefit from this practice.

Typical data augmentation techniques use a very limited set of known invariances that are easy to invoke. In this paper we recognize that we can learn a model of a much larger invariance space, through training a form of conditional generative adversarial network (GAN) in a different domain, typically called the *source domain*. This can then be applied in the low-data domain of interest, the

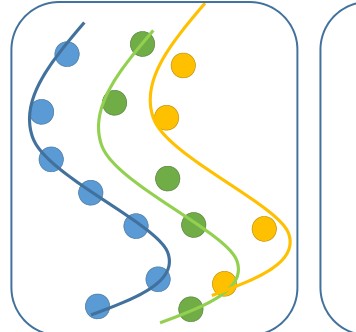 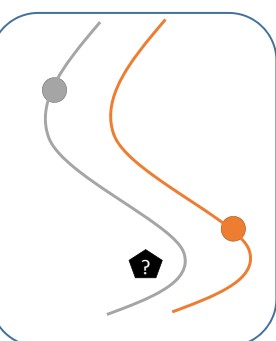

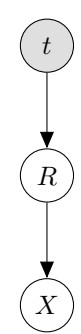

Figure 1: Learning a generative manifold for the classes in the source domain (left) can help learn better classifiers for the one shot target domain (right): The test point (pentagon) is nearer to the orange point but is actually closer to the learnt grey data manifold. If we generate extra examples on the grey manifold the nearest neighbour measure will better match the nearest manifold measure.

Figure 2: Graphical model for dataset shift in the one-shot setting: the distribution over class label $t$ changes in an extreme way, affecting the distribution over latent R. However the generating distribution $P(X|R)$ does not change.

*target domain*. We show that such a Data Augmentation Generative Adversarial Network (DAGAN) enables effective neural network training even in low-data target domains. As the DAGAN does not depend on the classes themselves, it captures the cross-class transformations, moving data-points to other points of equivalent class. As a result it can be applied to novel unseen classes. This is illustrated in Figure 2.

We also demonstrate that a learnt DAGAN can be used to substantially and efficiently improve Matching Networks (Vinyals et al., 2016) for few-shot target domains. It does this by augmenting the data in matching networks and related models (Vinyals et al., 2016; Snell et al., 2017) with the most relevant comparator points for each class generated from the DAGAN. This relates to the idea of tangent distances (Simard et al., 1991), but involves targeting the distances between manifolds defined by a learnt DAGAN. See Figure 1.

In this paper we train a DAGAN and then evaluate its performance on low-data target domains using (a) standard stochastic-gradient neural network training, and (b) specific one-shot meta-learning[1] methods. We use 3 datasets, the Omniglot dataset, the EMNIST dataset and the more complex VGG-Face dataset. The DAGAN trained from Omniglot was used for the unseen Omniglot terget domain, but was also for the EMNIST domain to demonstrate benefit even when transferring between substantially different source and target domains. The VGG-Face dataset provide a substantially more challenging test. We present generated samples which are of good quality, and also show substantial improvement on classification tasks via vanilla networks and matching networks.

The primary contributions of this paper are:

1. Using a novel Generative Adversarial Network to learn a representation and process for a data augmentation.

2. Demonstrate realistic data-augmentation samples from a single novel data point.

3. The application of DAGAN to augment a standard classifier in the low-data regime, demonstrating significant improvements in the generalization performance on all tasks.

4. The application of DAGAN in the meta-learning space, showing performance better than all previous general meta-learning models. The results showed performance beyond state-of-the art by 0.5% in the Omniglot case and 1.8% in the EMNIST case.

5. Efficient one shot augmentation of matching networks by learning a network to generate only the most salient augmentation examples for any given test case.

To our knowledge, this is the first paper to demonstrate state-of-the-art performance on meta-learning via novel data augmentation strategies.

## 2 BACKGROUND

**Transfer Learning and Dataset Shift:** the one shot learning problem is a particular case of dataset shift, summarised by the graphical model in Figure 2. The term *dataset shift* (Storkey, 2009)

---

[1]By meta-learning, we mean methods that learn a particular approach from a source domain and then use that approach in a different target domain.

generalises the concept of covariate shift (Shimodaira, 2000; Storkey & Sugiyama, 2007; Sugiyama & Müller, 2005) to multiple cases of changes between domains. For one shot learning there is an extreme shift in the class distributions - the two distributions share no support: the old classes have zero probability and the new ones move from zero to non-zero probability. However there is an assumption that the class conditional distributions share some commonality and so information can be transferred from the source domain to the one-shot target domain.

**Generative Adversarial Networks** (GAN) (Goodfellow et al., 2014), and specifically Deep Convolutional GANs (DCGAN) (Radford et al., 2015) use of the ability to discriminate between true and generated examples as an objective, GAN approaches can learn complex joint densities. Recent improvements in the optimization process (Arjovsky et al., 2017; Gulrajani et al., 2017) have reduced some of the failure modes of the GAN learning process.

**Data Augmentation** (Krizhevsky et al., 2012) is routinely used in classification problems. Often it is non-trivial to encode known invariances in a model. It can be easier to encode those invariances in the data instead by generating additional data items through transformations from existing data items. For example the labels of handwritten characters should be invariant to small shifts in location, small rotations or shears, changes in intensity, changes in stroke thickness, changes in size etc. Almost all cases of data augmentation are from a priori known invariance. Prior to this paper we know of few works that attempt to learn data augmentation strategies. One paper that is worthy of note is the work of (Hauberg et al., 2016), where the authors learn augmentation strategies on a class by class basis. This approach does not transfer to the one-shot setting where completely new classes are considered.

**Few-Shot Learning and Meta-Learning:** there have been a number of approaches to few shot learning, from (Salakhutdinov et al., 2012) where they use a hierarchical Boltzmann machine, through to modern deep learning architectures for one-shot conditional generation in (Rezende et al., 2016), hierarchical variational autoencoders in (Edwards & Storkey, 2017) and most recently a GAN based one-shot generative model (Mehrotra & Dukkipati, 2017). One early but effective approach to one-shot learning involved the use of Siamese networks (Koch, 2015). Others have used nonparametric Bayesian approaches (Salakhutdinov et al., 2012), and conditional variational autoencoders (Rezende et al., 2016). With few examples a nearest neighbour classifier or kernel classifier is an obvious choice. Hence meta-learning distance metrics or weighting kernels has clear potential (Vinyals et al., 2016; Snell et al., 2017). Skip-residual pairwise networks have also proven particularly effective (Mehrotra & Dukkipati, 2017). Various forms of memory augmented networks have also been used to collate the critical sparse information in an incremental way (Santoro et al., 2016; Vinyals et al., 2016). None of these approaches consider an augmentation model as the basis for meta-learning.

## 3 MODELS FOR DATA AUGMENTATION

If we know that a class label should be invariant to a particular transformation then we can apply that transformation to generate additional data. If we do not know what transformations might be valid, but we have other data from related problems, we can attempt to learn valid transformations from those related problems that we can apply to our setting (Figure 1). This is an example of meta-learning; we learn on other problems how to improve learning for our target problem. This is the subject of this paper.

Generative Adversarial Methods (Goodfellow et al., 2014) are one approach for learning to generate examples from density matching the density of a training dataset $D = \{\mathbf{x}_1, \mathbf{x}_2, \ldots, \mathbf{x}_{N_D}\}$ of size denoted by $N_D$. They learn by minimizing a distribution discrepancy measure between the generated data and the true data. The generative model learnt by a Generative Adversarial Network (GAN) takes the form[2]

$$\mathbf{z} = \tilde{N}(\mathbf{0}, \mathbf{I}) \tag{1}$$
$$\mathbf{v} = \mathbf{f}(\mathbf{z}) \tag{2}$$

where $\mathbf{f}$ is implemented via a neural network. Here, $\mathbf{v}$ are the vectors being generated (that, in distribution, should match the data $D$), and $\mathbf{z}$ are the latent Gaussian variables that provide the variation in what is generated.

---

[2]The assumption of a Gaussian $\mathbf{z}$ is without much loss of generality - all the other usual generating distribution are just known nonlinear transformations from a Gaussian; transformations that are fairly straightforward to implement via a neural network.

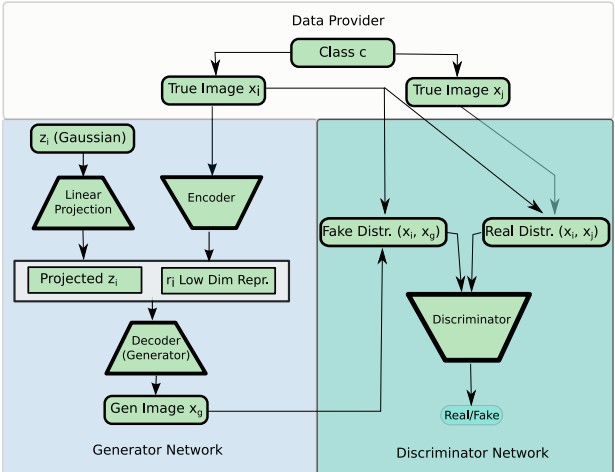

Figure 3: DAGAN Architecture. Left: the generator network is composed of an encoder taking an input image (from class c), projecting it down to a lower dimensional manifold (bottleneck). A random vector ($z_i$) is transformed and concatenated with the bottleneck vector; these are both passed to the decoder network which generates an augmentation image. Right: the adversarial discriminator network is trained to discriminate between the samples from the *real* distribution (other real images from the same class) and the *fake* distribution (images generative from the generator network). Adversarial training leads the network to generate new images from an old one that appear to be within the same class (whatever that class is), but look different enough to be a different sample.

A generative adversarial network can be thought of as learning transformations that map out a data manifold: $\mathbf{z} = \mathbf{0}$ gives one point on the manifold, and changing $\mathbf{z}$ in each different direction continuously maps out the data manifold.

### 3.1 DATA AUGMENTATION GENERATIVE ADVERSARIAL NETWORK

A generative adversarial network could also be used to map out a data augmentation manifold. Given a data point $\mathbf{x}$ we could learn a representation of the input $\mathbf{r} = \mathbf{g}(\mathbf{x})$, along with a generative model as before. The combined model would take the form

$$
\begin{aligned}
\mathbf{r} &= \mathbf{g}(\mathbf{x}) \\
\mathbf{z} &= \tilde{N}(\mathbf{0}, \mathbf{I}) \\
\mathbf{x} &= \mathbf{f}(\mathbf{z}, \mathbf{r})
\end{aligned}
\tag{3}
$$

where the neural network $\mathbf{f}$ now takes the representation $\mathbf{r}$ and the random $\mathbf{z}$ as inputs. Now given any new $\mathbf{x}^*$ we can

- Obtain a generatively meaningful representation of the data point $\mathbf{r}^* = g(\mathbf{x}^*)$, that encapsulates the information needed to generate other related data.
- Generate extra augmentation data $\mathbf{v}_1^*, \mathbf{v}_2^*, \ldots$ that supplements the original $\mathbf{x}^*$, which can be used in a classifier. This can be done y sampling $\mathbf{z}$ from the standard Gaussian distribution and then using the generative network to generate an augmentation example.

The precise form of data augmentation model we use in this paper is given in Figure 3. An outline of the structure of the model is elaborated in the figure caption.

### 3.2 LEARNING

The data augmentation model (3) can be learnt in the source domain using an adversarial approach. Consider a source domain consisting of data $D = \{\mathbf{x}_1, \mathbf{x}_2, \ldots, \mathbf{x}_{N_D}\}$ and corresponding target values $\{t_1, t_2, \ldots, t_{N_D}\}$. We use an improved WGAN (Gulrajani et al., 2017) critic that either takes

**(a)** some input data point $\mathbf{x}_i$ and a second data point from the same class: $\mathbf{x}_j$ such that $t_i = t_j$.

**(b)** some input data point $\mathbf{x}_i$ and the output of the current generator $\mathbf{x}_g$ which takes $\mathbf{x}_i$ as an input.

The critic tries to discriminate the generated points (b) from the real points (a). The generator is trained to minimize this discriminative capability as measured by the Wasserstein distance (Arjovsky et al., 2017).

The importance of providing the original **x** to the discriminator should be emphasised. We want to ensure the generator is capable of generating different data that is related to, but different from, the current data point. By providing information about the current data point to the discriminator we prevent the GAN from simply autoencoding the current data point. At the same time we do not provide class information, so it has to learn to generalise in ways that are consistent across all classes.

## 4 ARCHITECTURES

In the main experiments, we used a DAGAN generator that was a combination of a UNet and ResNet, which we henceforth call a UResNet. The UResNet generator has a total of 8 blocks, each block having 4 convolutional layers (with leaky rectified linear (relu) activations and batch renormalisation (batchrenorm) (Ioffe, 2017)) followed by one downscaling or upscaling layer. Downscaling layers (in blocks 1-4) were convolutions with stride 2 followed by leaky relu, batch normalisation and dropout. Upscaling layers were stride 1/2 replicators, followed by a convolution, leaky relu, batch renormalisation and dropout. For Omniglot and EMNIST experiments, all layers had 64 filters. For the VGG-Faces the first 2 blocks of the encoder and the last 2 blocks of the decoder had 64 filters and the last 2 blocks of the encoder and the first 2 blocks of the decoder 128 filters.

In addition each block of the UResNet generator had skip connections. As with a standard ResNet, a strided 1x1 convolution also passes information between blocks, bypassing the between block non-linearity to help gradient flow. Finally skip connections were introduced between equivalent sized filters at each end of the network (as with UNet). A figure of the architecture and a pseudocode definition is given in Appendix A.

We used a DenseNet (Huang et al., 2016) discriminator, using layer normalization instead of batch normalization; the latter would break the assumptions of the WGAN objective function. The DenseNet was composed of 4 Dense Blocks and 4 Transition Layers, as they were defined in (Huang et al., 2016). We used a growth rate of $k = 64$ and each Dense Block had 4 convolutional layers within it. For the discriminator we also used dropout at the last convolutional layer of each Dense Block as we found that this improves sample quality.

We trained each DAGAN for 500 epochs, using a learning rate of 0.0001, and an Adam optimizer with Adam parameters of $\beta_1 = 0$ and $\beta_2 = 0.9$.

For each classification experiment we used a DenseNet classifier composed of 4 Dense Blocks and 4 Transition Layers with a $k = 64$, each Dense Block had 3 convolutional layers within it. The classifiers were a total of 17 layers (i.e. 16 layers and 1 softmax layer). Furthermore we applied a dropout of 0.5 on the last convolutional layer in each Dense Block. The classifier was trained with standard augmentation: random Gaussian noise was added to images (with 50% probability), random shifts along x and y axis (with 50% probability), and random 90 degree rotations (all with equal probability of being chosen). Classifiers were trained for 200 epochs, a learning rate of 0.001, and an Adam optmizer with $\beta_1 = 0.9$ and $\beta_2 = 0.99$.

## 5 DATASETS

We tested the DAGAN augmentation on 3 datasets: Omniglot, EMNIST, and VGG-Faces. All datasets were split randomly into source domain sets, validation domain sets and test domain sets.

For classifier networks, all data for each character (handwritten or person) was further split into 2 test cases (for all datasets), 3 validation cases and a varying number of training cases depending on the experiment. Classifier training was done on the training cases for all examples in all domains, with hyperparameter choice made on the validation cases. Finally test performance was reported only on the test cases for the target domain set. Case splits were randomized across each test run.

For one-shot networks, DAGAN training was done on the source domain, and the meta learning done on the source domain, and validated on the validation domain. Results were presented on the target domain data. Again in the target domain a varying number of training cases were provided and results were presented on the test cases (2 cases for each target domain class in all datasets).

The Omniglot data (Lake et al., 2015) was split into source domain and target domain similarly to the split in (Vinyals et al., 2016). The order of the classes was shuffled such that the source and target

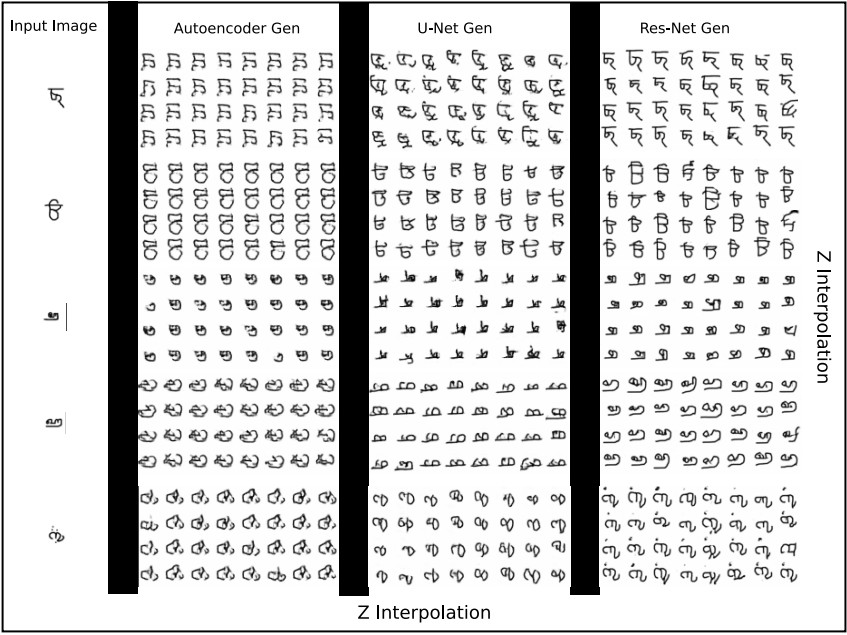

Figure 4: Omniglot DAGAN generations with different architectures.

domains contain diverse samples (i.e. from various alphabets). The first 1200 were used as a source domain set, 1201-1412 as a validation domain set and 1412-1623 as a target domain test set.

The EMNIST data was split into a source domain that included classes 0-34 (after random shuffling of the classes), the validation domain set included classes 35-42 and the test domain set included classes 42-47. Since the EMNIST dataset has thousands of samples per class we chose only a subset of 100 for each class, so that we could make our task a low-data one.

In the VGG-Face dataset case, we randomly chose 100 samples from each class that had 100 uncorrupted images, resulting in 2396 of the full 2622 classes available in the dataset. After shuffling, we split the resulting dataset into a source domain that included the first 1802 classes. The test domain set included classes 1803-2300 and the validation domain set included classes 2300-2396.

## 6 DAGAN TRAINING AND GENERATION

### 6.1 TRAINING OF DAGAN IN SOURCE DOMAIN

A DAGAN was trained on Source Omniglot domains using a variety of architectures: standard VGG, U-nets, and ResNet inspired architectures. Increasingly powerful networks proved better generators, with the UResNet described in Section 4 generator being our model of choice. Figure 4 shows the improved variability in generations that is achieved with more powerful architectures. The DAGAN was also trained on the VGG-Faces source domains. Examples of the generated faces are given in Figure 5

### 6.2 VANILLA CLASSIFIERS

The first test is how well the DAGAN can augment vanilla classifiers trained on each target domain. A DenseNet classifier (as described in Section 4) was trained first on just real data (with standard data augmentation) with 5, 10 or 15 examples per class. In the second case, the classifier was also passed DAGAN generated augmented data. The real or fake label was also passed to the network, to enable the network to learn how best to emphasise true over generated data. This last step proved crucial to maximizing the potential of the DAGAN augmentations. In each training cycle, varying numbers of augmented samples were provided for each real example (ranging from 1-10); the best annotation rate was selected via performance on the validation domain. The results on the held out test cases from the target domain is given in Table 1. In every case the augmentation improves the classification.

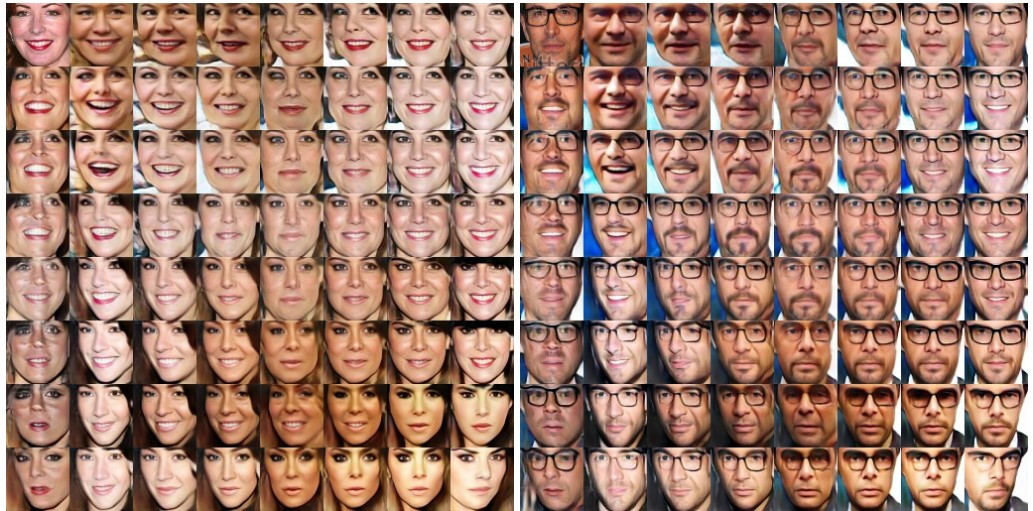

Figure 5: An Interpolated spherical subspace of the GAN generation space using a single real seed image (top left corner). The only real image in each figure is the one in the top-left corner, the rest are generated to augment that example using a DAGAN.

.

| Omniglot DAGAN Augmented Classification | | |
|---|---|---|
| **Experiment ID** | **Samples Per Class** | **Test Accuracy** |
| Omni_5_Standard | 5 | 0.689904 |
| Omni_5_DAGAN_Augmented | 5 | **0.821314** |
| Omni_10_Standard | 10 | 0.794071 |
| Omni_10_DAGAN_Augmented | 10 | **0.862179** |
| Omni_15_Standard | 15 | 0.819712 |
| Omni_15_DAGAN_Augmented | 15 | **0.874199** |
| **EMNIST DAGAN Augmented Classification** | | |
| **Experiment ID** | **Samples Per Class** | **Test Accuracy** |
| EMNIST_Standard | 15 | 0.739353 |
| EMNIST_DAGAN_Augmented | 15 | **0.760701** |
| EMNIST_Standard | 25 | 0.783539 |
| EMNIST_DAGAN_Augmented | 25 | **0.802598** |
| EMNIST_Standard | 50 | 0.815055 |
| EMNIST_DAGAN_Augmented | 50 | **0.827832** |
| EMNIST_Standard | 100 | 0.837787 |
| EMNIST_DAGAN_Augmented | 100 | **0.848009** |
| **Face DAGAN Augmented Classification** | | |
| **Experiment ID** | **Samples Per Class** | **Test Accuracy** |
| VGG-Face_Standard | 5 | 0.0446948 |
| VGG-Face_DAGAN_Augmented | 5 | **0.125969** |
| VGG-Face_Standard | 15 | 0.39329 |
| VGG-Face_DAGAN_Augmented | 15 | **0.429385** |
| VGG-Face_Standard | 25 | 0.579942 |
| VGG-Face_DAGAN_Augmented | 25 | **0.584666** |

Table 1: Vanilla Classification Results: All results are averages over 5 independent runs. The DAGAN augmentation improves the classifier performance in all cases. Test accuracy is the result on the test cases in the test domain

# 7 ONE SHOT LEARNING USING DATA AUGMENTATION NETWORKS AND MATCHING NETWORKS

A standard approach to one shot learning involves learning an appropriate distance between representations that can be used with a nearest neighbour classifier under some metric. We focus on the use of Matching Networks to learn a representation space, where distances in that representation space produce good classifiers. The same networks can then be used in a target domain for nearest neighbour classification.

Matching networks were introduced by (Vinyals et al., 2016). A matching network creates a predictor from a support set in the target domain by using an attention memory network to generate an appropriate comparison space for comparing a test example with each of the training examples. This is achieved by simulating the process of having small support sets from the source domain and learning to learn a good mapping.

However matching networks can only learn to match based on individual examples. By augmenting the support sets and then learning to learn from that augmented data, we can enhance the classification power of matching networks, and apply that to the augmented domain. Beyond that we can learn to choose good examples from the DAGAN space that can best related the test and training manifolds. Precisely, we train the DAGAN on the source domain, then train the matching networks on the source domain, along with a *sample-selector* neural network that selects the best representative **z** input used to create an additional datum that augments each case. As all these quantities are differentiable, this whole process can be trained as a full pipeline. Networks are tested on the validation domain, and the best performing matching network and sample-selector network combination are selected for use in the test domain.

When training matching networks with DAGAN augmentation, augmentation was used during every matching network training episode to simulate the data augmentation process. We used matching networks without full-context embedding version and stacked K GAN generated (augmented) images along with the original image. We ran experiments for 20 classes and one sample per class per episode, i.e. the one shot learning setting. The ratio of generated to real data was varied from 0 to 2. Once again standard augmentations were applied to all data in both DAGAN augmented and standard settings. In Table 2, showing the Omniglot one-shot results, we see that the DAGAN was enhancing even simple pixel distance with an improvement of 33.815%. Furthermore in our matching network experiments we saw an improvement of 0.5% over the state of the art (96.9% to 97.4%), pushes the matching network performance to the level of the Conv Arc (Shyam et al., 2017) technique which used substantial prior knowledge of the data form (that the data was built from pen strokes) to achieve those results.

In addition to Omniglot, we experimented on EMNIST and VGG-Face. We used the Omniglot DAGAN to generate augmentation samples for EMNIST to stress test the power of the network to generalise over dissimilar domains. The one-shot results showed an improvement of 1.8% over the baseline matching network that was training on EMNIST (i.e. from 59.5% to 61.3%, again averaged over 3 runs). In the VGG-Face one shot matching networks experiments we saw that the the DAGAN augmented system performed with same performance as the baseline one. However it is worth noting that the non augmented matching network did not overfit the dataset as it happened in Omniglot and EMNIST experiments. This indicates that the embedding network architecture we used perhaps did not have enough capacity to learn a complex task such as discriminating between faces. The embedding network architecture used was the one described in (Vinyals et al., 2016) with 4 convolutional layers with 64 filters each. Further experiments with larger embedding functions are required to better explore and evaluate the DAGAN performance on VGG-Face dataset.

# 8 CONCLUSIONS

Data augmentation is a widely applicable approach to improving performance in low-data setting, and a DAGAN is a flexible model to automatic learn to augment data. However beyond that, We demonstrate that DAGANS improve performance of classifiers even after standard data-augmentation. Furthermore by meta-learning the best choice of augmentation in a one-shot setting it leads to better performance than other state of the art meta learning methods. The generality of data augmentation across all models and methods means that a DAGAN could be a valuable addition to any low data setting.

| Technique Name | Test Accuracy |
|---|---|
| Pixel Distance | 0.267 |
| **Pixel Distance + DAGAN Augmentations** | **0.60515** |
| Matching Nets | 0.938 |
| Neural Statistician | 0.931 |
| **Conv. ARC** | **0.975** |
| Prototypical Networks | 0.96 |
| Siam-I | 0.884 |
| Siam-II | 0.92 |
| GR + Siam-I | 0.936 |
| GR + Siam-II | 0.912 |
| SRPN | 0.948 |
| **Matching Nets (Local Reproduction)** | **0.969** |
| **Matching Nets + DAGAN Augmentations** | **0.974** |

Table 2: Omniglot One Shot Results: All results are averages over 3 independent runs. Note that our own local implementation of matching networks substantially outperform the matching network results presented in the original paper, However DAGAN augmentation takes matching networks up to the level of Conv-ARC (Shyam et al., 2017), which explicitly use knowledge that the data has the structure of handwritten characters: the Conv-ARC model uses the stroke structure of the characters to perform well. Note DAGAN augmentation can even increase a simple pixel distance nearest neighbour model up to non-negligible levels.

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

## A  APPENDIX

In this Appendix we show the form of the UResNet generator network in Figure 6 and the full details of the UResNet model in Algorithm 1. Full code implementing all aspects of this paper will be made available on acceptance.

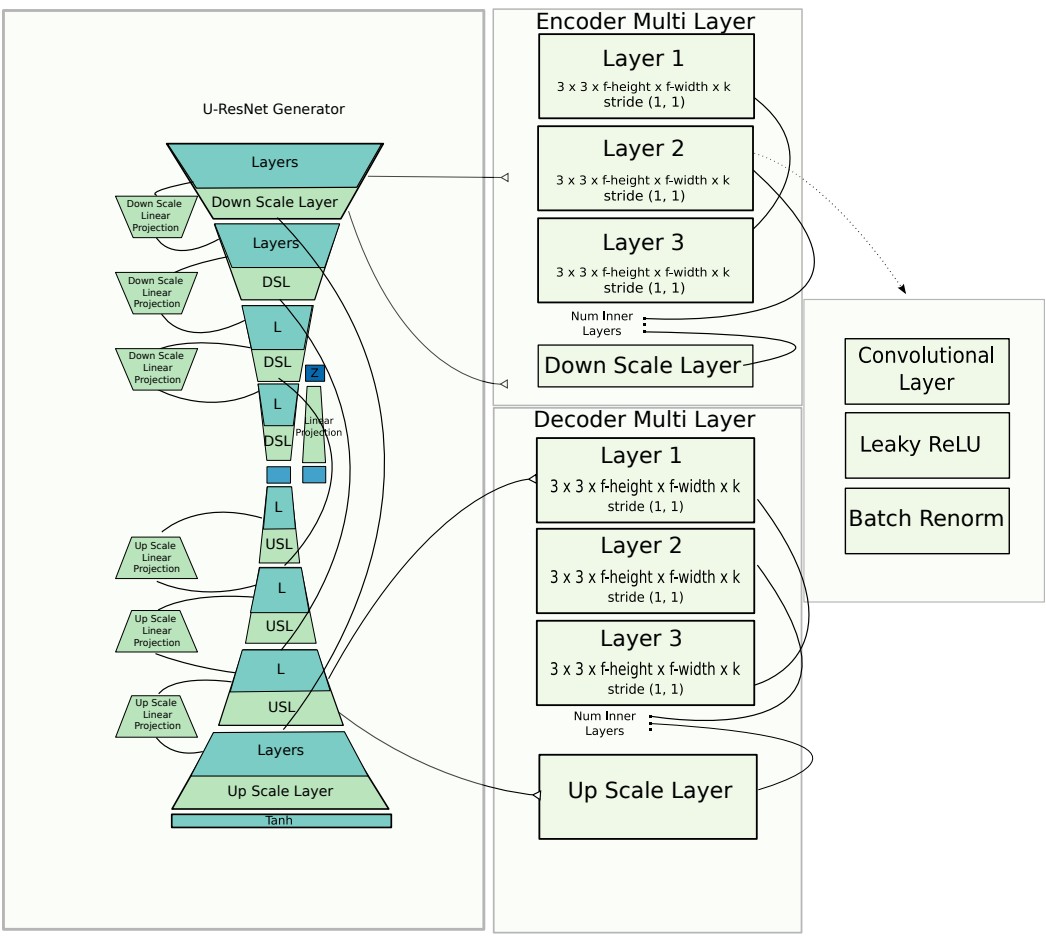

Figure 6: UResNet Generator: In this figure one can see a drawing of the UResNet generator as described in Algorithm 1.

---

**Algorithm 1** U-ResNet Generator Architecture

---

1: The notation is as follows:
2: EML() ⇒ Encoder_Multi_Layer()
3: DML() ⇒ Decoder_Multi_Layer()
4: L() ⇒ Layer()
5: ML() ⇒ Multi_Layer()
6: USL() ⇒ Up_Scale_Layer()
7: DSL() ⇒ Down_Scale_Layer()
8: USLP() ⇒ Up_Scale_Linear_Projection()
9: DSLP() ⇒ Down_Scale_Linear_Projection()
10: **procedure** U-RESNET(x, l, n, k_list, skip_distance)
11:    $p = None$
12:    **for** $i \in \{1, \ldots, l-1\}$ **do**
13:        $x, p \leftarrow EML(x, n, skip\_distance, k = k\_list[i], p = p)$
14:    $z \leftarrow$ *Sample 100-dim N(0, 1)*
15:    *project z to match dimensionality of x*
16:    $x = concat(x, z)$
17:    $p = None$
18:    **for** $i \in \{1, \ldots, l-1\}$ **do**
19:        $x, p \leftarrow DML(x, n, skip\_distance, k = k\_list[i], p = p)$
20: **procedure** ENCODER_MULTI_LAYER(x, n, skip_distance, k, p=None)
21:    $x \leftarrow ML(x, n, skip\_distance, k, p = p)$
22:    $p \leftarrow DSLP(x, k)$
23:    $x \leftarrow DSL(x, k)$
24:    **return x, p**
25: **procedure** DECODER_MULTI_LAYER(x, n, skip_distance, k, p=None)
26:    $x \leftarrow ML(x, n, skip\_distance, k, p = p)$
27:    $p \leftarrow USLP(x, k)$
28:    $x \leftarrow USL(x, k)$
29:    **return x, p**
30: **procedure** MULTI_LAYER(x, n, skip_distance, k, p=None)
31:    **if** $p == None$ **then** $prev \leftarrow list([x])$
32:    **else** $prev \leftarrow list([concat(x, p)])$
33:    **for** $i \in \{1, \ldots, n-1\}$ **do**
34:        $x \leftarrow concat(prev)$
35:        $x \leftarrow L(x, k, s = 1)$
36:        **if** $len(prev) >= skip\_distance$ **then** prev = list([x])
37:        **else** $prev.append(x)$
38:    **return x**
39: **procedure** DOWN_SCALE_LAYER(x, k)
40:    $x \leftarrow$ L(x, k, s=2)
41:    $x \leftarrow$ Dropout(x, 0.3)
42:    **return x**
43: **procedure** DOWN_SCALE_LINEAR_PROJECTION(x, k)
44:    $x \leftarrow$ L(x, fsize=1, k=k, s=2)
45:    **return x**
46: **procedure** UP_SCALE_LAYER(x, k)
47:    $x \leftarrow$ nearest_neighbour_resize(x, s=2)
48:    $x \leftarrow$ L(x, k, s=1)
49:    **return x**
50: **procedure** UP_SCALE_LINEAR_PROJECTION(x, k)
51:    $x \leftarrow$ nearest_neighbour_resize(x, s=2)
52:    $x \leftarrow$ Conv2D(x, fsize=1, s=1, k=k)
53:    **return x**

---

