# OpenReview forum: "Data Augmentation Generative Adversarial Networks"
_ICLR.cc/2018/Conference — Invite to Workshop Track_

### Official Review · AnonReviewer1 · 2017-11-26
**This paper considers the data-augmentation problem which is very interesting. However, I don't see enough contribution in the current version.**

**Rating:** 4
**Confidence:** 4

**Review:**

This paper proposes a conditional Generative Adversarial Networks that is used for data augmentation. In order to evaluate the performance of the proposed model, they use Omniglot, EMNIST, and VGG-Faces datasets and uses in the meta-learning task and standard classification task in the low-data regime. The paper is well-written and consistent.

Even though this paper learns to do data-augmentation (which is very interesting ) rather than just simply applies some standard data augmentation techniques and shows improvements in some tasks, I am not convinced about novelty and originality of this paper, especially on the model side. To be more specific, the paper uses the previously proposed conditional GAN as the main component of their model. And for the one-shot learning tasks, it only trains the previously proposed models with these newly augmented data.

In addition, there are some other works that used GAN as a method for some version of data augmentation:
- RenderGAN: Generating Realistic Labeled Data
  https://arxiv.org/abs/1611.01331
-Data Augmentation in Emotion Classification Using Generative Adversarial Networks
https://arxiv.org/abs/1711.00648

It is fair to say that their model shows improvement on the above tasks but this improvement comes with a cost of training of GAN network.

In summary, the idea of the paper is very interesting to learn data-augmentation but yet I am not convinced the current paper has enough novelty and contribution and see the contribution of paper as on more the application side rather than on model and problem side. That said I'd be happy to hear the argument of the author about my comments.

---

> ### Author Response · Authors · 2017-12-14
> **Elaborations on review concerns and many thanks**
>
> Thanks for your review and time. I will address your concerns in sections:
>
> Model Novelty:
>
> The model is not a standard conditional GAN as it’s actually image conditioned and not label conditioned as per RenderGAN. A DAGAN is attempting to meta-learn to one-shot generate plausible versions of a provided image, which are varied by the random injected noise. To do so we use a novel training scheme/setup which is where the key contribution of the work lies. By allowing the Generator to learn classes implicitly rather than explicitly we open the possibility for the DAGAN to one shot generate samples on previously unseen classes. In the one shot case, we don’t just augment the training set, we are actually producing on the fly generated samples conditioned on unseen classes in training, validation and test times. By doing so we are converting the one-shot setup, to a few shot setup.
>
> Further Novel Contributions:
>
> When we generate samples on the fly for the matching network we also provide the matching network with information on the source of the images (i.e. real/fake) by doing so the network can learn how much trust to put in fake augmented examples, and adjust the  embedding based upon the real/fake label, which improves accuracy performance. In addition we also learn a network that given the target image for a certain episode can generate the best Z for that specific task, which again improves performance.
>
> Architectural Contributions:
>
> We have built a novel generator architecture which combines ideas from ResNets, DenseNets and U-Nets to generate very high quality results. Furthermore we use batch renormalization which in our empirical evaluation is shown to greatly enhance sample quality and sample variation, therefore providing evidence for some of the theoretical claims made in the original Batch Renormalization paper. https://arxiv.org/abs/1702.03275
>
> Flexibility of method:
>
> A DAGAN is compatible with any few-shot learning technique so that as new few-shot learning ideas are created, DAGANs can be used to squeeze out extra information from the data thus building more data efficient systems. In addition DAGAN can be further improved by future advances in training GANs.
>
> Also to summarise our improvements over the mentioned papers:
>
> RenderGAN uses label conditioned GANs which cannot be used for one-shot generation on unseen classes. DAGAN can do one-shot generation on unseen classes as it learns the concept of a class implicitly and is conditioned on an image, rather than a label.
> Data Augmentation in Emotion Classification Using GAN: Here the authors use a CycleGAN which requires 2 Generators, 2 Discriminators and a rather complicated amount of loss functions to train. Our model only requires 1 Generator and 1 Discriminator and the GAN Loss. This makes it far less computationally expensive and also produces results that at least visually appear to be much higher quality. In addition in their paper they do not mention whether the model can one-shot generate good samples from unseen classes which is something DAGAN does very well.
>
> As far as the cost of training the GAN, a UResNet grade Omniglot network needs about 12 hours on a single Titan X Pascal and was then used to run one-shot generation on both Omniglot and EMNIST (thus showing how well the DAGAN can generate samples from unseen classes or in this case, unseen datasets). Yes, it requires additional computational overhead but as demonstrated the improvements are well worth it.
>
> I’d be more than happy to discuss any other concerns you might have and I will make a good attempt to improve the clarity of the illustration of the network and emphasize our contributions. Once again, thank you for your review, looking forward to your reply.

---

### Official Review · AnonReviewer3 · 2017-11-27
**The proposition is technically sound and the novelty is significant. However, the illustration is not clear enough and need improving.**

**Rating:** 9
**Confidence:** 5

**Review:**

In this paper, the authors have proposed a GAN based method to conduct data augmentation. The cross-class transformations are mapped to a low dimensional latent space using conditional GAN. The paper is technically sound and the novelty is significant. The motivation of the proposed methods is clearly illustrated. Experiments on three datasets demonstrate the advantage of the proposed framework. However, this paper still suffers from some drawbacks as below:
(1)	The illustration of the framework is not clear enough. For example, in figure 3, it says the GAN is designed for “class c”, which is ambiguous whether the authors trained only one network for all class or trained multiple networks and each is trained on one class.
(2)	Some details is not clearly given, such as the dimension of the Gaussian distribution, the dimension of the projected  noise and .
(3)	The proposed method needs to sample image pairs in each class. As far as I am concerned, in most cases sampling strategy will affect the performance to some extent. The authors need to show the robustness to sampling strategy of the proposed method.

---

> ### Author Response · Authors · 2017-12-14
> **Elaboration on drawbacks and many thanks**
>
> Thanks for your review and time. I am very glad you like our work. I will address your concerns using the same identifiers you have used.
>
> I agree with your observation, it seems to be a common issue in all 3 reviews that we need a better illustration/more textual description in Figure 3. Furthermore, we trained 1 DAGAN for all classes. This is key in fact, since we condition the GAN on an image from the class we want to generate from. Thus training for all classes allows the generator to learn augmentations from all the classes and apply them to different classes in a way that allows the samples to remain within their original class, therefore leveraging our data more efficiently.
> The dimension of the Gaussian is 100-dimensional. The projected noise dimensionality is different from dataset to dataset depending on the image dimensionality. In all cases we make sure that the projected noise matches the size of the encoder embedding.
> Yes, the sampling strategy is perhaps one of the most important parts of our methodology. When constructing a new training sample we choose 1 class and then 2 samples from that class using a uniform distribution to use for x_i and x_j whilst making sure the 2 samples are different samples and not identical. This way we are providing the network with 2 unique samples that are always varied at each iteration. There is no label information provided to the DAGAN as we want the Generator to learn to one-shot generate samples that are within the same class of the conditional image, thus pushing the Generator to implicitly learn a manifold around a data sample within which the sample remains in the same class but is augmented enough to be a different sample than the conditional one. The augmentations learned are learned from the whole dataset and often we see the transfer of augmentations from one class to another, only where it makes sense (i.e. add lipstick to females but not males).
> Once again, thanks for your review and time. I’d be more than happy to discuss any other concerns you might have.

---

### Official Review · AnonReviewer2 · 2017-11-28
**This paper is good at using the GAN for data augmentation for the one shot learning, and have demonstrated good performance for a variety of datasets.**

**Rating:** 6
**Confidence:** 3

**Review:**

This paper is good at using the GAN for data augmentation for the one shot learning, and have demonstrated good performance for a variety of datasets.
However, it seems that the main technique contribution is not so clear. E.g., it is not clear as shown in Figure 3, what is key novelty of the proposed DAGAN, and how does it improve from the existing GAN work. It seems that the paper is a pipeline of many existing works.
Besides, it will also be interested to see whether this DAGAN can help in the training of prevailing ImageNet and MS COCO tasks.

---

> ### Author Response · Authors · 2017-12-14
> **Further Elaborations on the requested matters**
>
> Thanks for your review and time. The key contribution of the paper is a new GAN training setup with which one can use existing GAN framework  (i.e. WGAN GP or Standard GAN) to learn to one-shot generate plausible interpolations of data samples. Intuitively the model learns a manifold around a data point within which a sample remains in the same class. Furthermore, the concept of class is extracted directly from the image pairs passed to the discriminator and implicitly learned by the Generator network as a result of backpropagation. One of the novelties of this Data Augmentation technique using GANs is that at generation time you are not restricted by the classes you have already learned (i.e. No Labels are passed to the generator) rather the generator can one-shot generate from unseen-class data points which is where the true power of the DAGAN lies. In fact when training we take note of the WGAN validation loss such that we do not overfit that measure. This allows the network to be used not only for data augmentation in classification but also in the few-shot learning scheme. In terms of attempting experiments on ImageNet and MS COCO we were unfortunately computationally constrained and thus unable to run those experiments within a suitable time frame.

---

### Decision · Program_Chairs · 2018-01-29
**ICLR 2018 Conference Acceptance Decision**

**Decision:**

Invite to Workshop Track

**Comment:**

The paper based on cGAN developed a data augmentation GAN to deal with unseen classes of data. The paper developed new modifications to each component and designed network structure using ideas from state-of-the-art nets. As pointed out by reviewer 1 & 2, the technical contribution is not sufficient. We hence recommend it to workshop publication.